# A Comprehensive Review on Source, Types, Effects, Nanotechnology, Detection, and Therapeutic Management of Reactive Carbonyl Species Associated with Various Chronic Diseases

**DOI:** 10.3390/antiox9111075

**Published:** 2020-11-02

**Authors:** Shivkanya Fuloria, Vetriselvan Subramaniyan, Sundram Karupiah, Usha Kumari, Kathiresan Sathasivam, Dhanalekshmi Unnikrishnan Meenakshi, Yuan Seng Wu, Rhanye Mac Guad, Kaviraja Udupa, Neeraj Kumar Fuloria

**Affiliations:** 1Faculty of Pharmacy, AIMST University, Kedah, Bedong 08100, Malaysia; sundram@aimst.edu.my; 2Faculty of Medicine, Bioscience and Nursing, MAHSA University, Kuala Lumpur 42610, Malaysia; drvetriselvan@mahsa.edu.my (V.S.); sywu@mahsa.edu.my (Y.S.W.); 3Faculty of Medicine, AIMST University, Kedah, Bedong 08100, Malaysia; usha_harischandran@aimst.edu.my; 4Faculty of Applied Science, AIMST University, Kedah, Bedong 08100, Malaysia; skathir@aimst.edu.my; 5Department of Pharmacology, National University of Science and Technology, Bowshar Campus, Muscat PO 620 PC 130, Oman; dhanalekshmi@omc.edu.om; 6Faculty of Medicine and Health Science, University Malaysia Sabah, Kota Kinabalu 88400, Malaysia; rhanye@ums.edu.my; 7Department of Neurophysiology, NIMHANS, Bangalore 560029, India; kaviudupa.nimhans@nic.in

**Keywords:** reactive carbonyl species, scavengers, identification, validation, chronic diseases, treatment approach

## Abstract

Continuous oxidation of carbohydrates, lipids, and amino acids generate extremely reactive carbonyl species (RCS). Human body comprises some important RCS namely hexanal, acrolein, 4-hydroxy-2-nonenal, methylglyoxal, malondialdehyde, isolevuglandins, and 4-oxo-2- nonenal etc. These RCS damage important cellular components including proteins, nucleic acids, and lipids, which manifests cytotoxicity, mutagenicity, multitude of adducts and crosslinks that are connected to ageing and various chronic diseases like inflammatory disease, atherosclerosis, cerebral ischemia, diabetes, cancer, neurodegenerative diseases and cardiovascular disease. The constant prevalence of RCS in living cells suggests their importance in signal transduction and gene expression. Extensive knowledge of RCS properties, metabolism and relation with metabolic diseases would assist in development of effective approach to prevent numerous chronic diseases. Treatment approaches for RCS associated diseases involve endogenous RCS metabolizers, carbonyl metabolizing enzyme inducers, and RCS scavengers. Limited bioavailability and bio efficacy of RCS sequesters suggest importance of nanoparticles and nanocarriers. Identification of RCS and screening of compounds ability to sequester RCS employ several bioassays and analytical techniques. Present review describes in-depth study of RCS sources, types, properties, identification techniques, therapeutic approaches, nanocarriers, and their role in various diseases. This study will give an idea for therapeutic development to combat the RCS associated chronic diseases.

## 1. Introduction

The cellular metabolic process continuously produces reactive oxygen species (ROS) that are important for homeostasis in the human body [1]. The dysregulated metabolism manifests in ROS load that cascades into several chronic diseases by damaging various important macromolecules, such as: nucleic acids, amino acids (of proteins), carbohydrates and unsaturated fatty acids [2,3]. Oxidation of amino acids, lipids and carbohydrates generates several reactive carbonyl species (RCS) [3]. The electrophilic property of RCS makes them extremely reactive against nucleophilic groups of proteins, nucleic acids, amino phospho-lipids and other macromolecules [4]. This results in cytotoxicity, mutagenicity, malfunctioning, multitude of adducts and crosslinks that are connected to several chronic diseases like inflammatory disease, rheumatoid arthritis (RA), post-ischemic reoxygenation injury, atherosclerosis, cerebral ischemia, diabetes, cancer, pulmonary diseases, neurodegenerative diseases (NDs) and cardiovascular diseases (CVDs) [5,6,7,8,9]. At lower concentrations RCS exhibit beneficial effects in human body such as: glycolytic, anticancer, antiprotozoal, antibacterial, anti-viral and antifungal activities. So, it is a need to investigate the concentration of RCS using various analytical techniques in the human body [3,10]. Hence, to develop an effective approach to prevent numerous chronic diseases, it is utmost important to understand the RCS properties and link with RCS metabolism and metabolic diseases.

## 2. RCS Sources and Types

The living organism are identified to produce more than twenty exogeneous and endogenous RCS [11]. The chemical structure of RCS that are mostly found in the biological samples are displayed in Figure 1.

The exogeneous RCS like glyoxal, formaldehyde, crotonaldehyde, acrolein and acetone are generated from widespread industrial pollutants that easily enter human cells [12,13]. Other exogeneous sources of RCS include food additives, cigarette smoke and organic pharmaceutical products [14,15]. Facts also suggest endogenous production of RCS during enzymatical and non-enzymatical reactions in vivo. These nonenzymatic reactions such as: glycation, amino acids oxidation, and lipid peroxidation (LP) generates wide variety of intracellular unstable RCS [16,17,18,19,20,21].

### 2.1. RCS Generation via LP

Break down of free radical chain of polyunsaturated fatty acid (PUFA) in triglycerides, phospholipids, and cholesterol esters is an important characteristic of LP. The LP process generates a broad range of RCS in form of aldehydes, such as: hexanal, acrolein, crotonaldehyde (CA), 4-hydroxy-2-nonenal (4-HNE), glyoxal (GO), 4-oxo-*trans*-2-nonenal (ONE), *trans*-2-nonenal and malondialdehyde (MDA) [22]. Commonly, among total LP-mediated RCS (aldehydes) production, the contribution of MDA, hexanal and HNE is 70%, 15% and 5% respectively [23]. Evidence suggest acrolein (environmental pollutant) as low density lipoprotein (LDL) oxidation-mediated RCS [24]. However, in diseased state the LP process generates RCS in high concentration, such that many RCS could be utilized as biomarkers for oxidative damage and advancement of diseases [25]. Brain with Alzheimer’s disease (AD) exhibits high acrolein, crotonaldehyde-protein adduct, and HNE level. In fact, the acrolein modified proteins level may be utilized as marker for AD [26,27,28,29]. Evidences over high concentration of HNE-protein adducts in plasma (in vivo hypertension model), fibrotic plaques, and oxidized LDL suggests HNE function in atherosclerosis [30,31]. For example, serum exhibits high HNE-albumin adducts level in Type II diabetes [32].

### 2.2. RCS Generation via Glycoxidation

The reducing sugar may produce Schiff base on reaction with amino group of amino acids (like arginine and lysine) via Maillard reactions. This mediates numerous rearrangements and generates several advanced glycation end (AGE) products. Oxidation of AGE products generates several RCS in form of α-oxoaldehydes or dicarbonyls (like: glyoxal (GO), methyl glyoxal (MGO), 3-deoxy- glucosone (DGO), carboxymethyllysine (CML), glyoxal-lysine-dimer (GOLD) and methylglyoxal- lysine dimer (MOLD); and short-chain aldehydes (like: acetol, acrolein diacetyl and pyruvaldehyde) [33,34]. Glycoxidation is caused when high levels of oxidants couples with excess glucose (the common problem in diabetes) [9]. Glycolytic flux may increase MGO and GO level that may react with proteins (amino acids) and enhance AGE formation. This in turn may increase the α-oxoaldehyde (the glycoxidation product) that may aggravate numerous chronic diseases. For example, the high MGO or GO level causes protein carbonylation and protein aggregates formation, that contributes in cataract formation [35]. The MGO also contributes in vascular complications of diabetes, AD, amyloid plaques formation and neurofibrillary tangles [36].

The non-enzymatic glycation involves reducing carbohydrates like fructose and glucose [37,38,39]. This process may be ascribed to excess of carbohydrate (sweeteners) consumption in human diet or opposite effect in vivo [40,41,42]. Study suggest RCS and ROS connectivity in cytotoxic and defensive effects of reducing carbohydrates (fructose) [43]. Commonly, the carbohydrates involving enzymatic pathways generates MGO, GO, and 3-deoxyglucosone (DG) as RCS. For example, Figure 2a demonstrates production of RCS in form of 3-DG in polyol metabolic pathway of glucose [44,45]. The glycolysis pathway given in Figure 2b, demonstrates production of RCS in form of MGO [46]. Apart from it MGO are also generated from oxidation of ketone bodies [20]. Study suggest in vivo generation of different RCS from human phagocytes and neutrophils that generates α-hydroxy- and α,β-unsaturated aldehydes from hydroxy-amino acids using myelo-peroxidase-H_2_O_2_-chloride system [47]. Oxidation of amino acids such as glycine and threonine generate various RCS like MGO, succinyl acetone (SA) and aminoacetone (AA) [3].

### 2.3. Types and Properties of RCS 

To find an effective approach against RCS induced chronic diseases, it’s very important to understand the properties/reactivity of RCS. Following are some commonly found RCS and their properties.

#### 2.3.1. 4-Hydroxy-2-Nonenal (HNE)

The in vivo generation of HNE (α,β-unsaturated aldehydes) precursor from ω-6 PUFA involves several mechanisms [48]. Reaction of HNE with nucleophiles like thiols or imidazoles offers Michael adducts (MA) that successively undergoes secondary reaction and results in cyclic hemiacetals. However, the rate of MA formation with thiols is faster than imidazoles [2,23]. The HNE when reacts with primary amines offer hemiacetals (MA) and pyrrole adducts [49]. The HNE reaction with arginine also offers MA [50]. In comparison to HNE, the reactivity of isolevuglandins (IsoLG) towards bovine serum albumin (BSA) is 100 times faster [51]. Apart from reactivity towards amino acid residues, with phosphatidyl ethanolamine (phospholipids) HNE offers MA and pyrrole adducts [52]. Research suggest HNE to offer etheno-DNA adduct with DNA base and at least one fluorescent crosslink adduct with protein crosslinks [53,54].

#### 2.3.2. 4-Oxo-2-Nonenal (ONE)

The ONE is α,β-unsaturated 1,4-dicarbonyl compound that is produced from ω-6 PUFA (the HNE precursor). Replacement of 4-hydroxy group of HNE with carbonyl group forms ONE with dramatic changes in reactivity and biochemical properties. For instance, primary reaction of ONE with S-H group of thiol compounds (cellular component) forms MA, whereas residual 4-ketoaldehyde continues secondary reaction with primary amines to generate pyrrole adduct. This secondary reaction represents potential mechanism of ONE to generate crosslinks [55]. For example, with primary amines ONE exhibits quick reaction and produces keto-amide adduct with high stability even in absence of thiols [56]. Reports suggest ONE as most damaging RCS that is produced in vivo [2].

#### 2.3.3. Acrolein

Acrolein is produced in vivo from the oxidation of amino acids, polyamines, glycerol and carbohydrates. Cigarette smoke is also considered as an important source for acrolein [57]. Non-enzymatic reaction of acrolein with S-H group of thiols compounds like glutathione or GSH (the cellular component) to form MA is 100 time faster than HNE [23]. Acrolein reaction with GSH is commonly catalyzed by different glutathione transferases (GST), that appears as key mechanisms in detoxification of acrolein [58]. Such reaction reduces GSH level on exposure of cells towards high acrolein concentration and lead to cells death [59]. Apart from reaction with cysteine (thiol group), acrolein also reacts with histidine and primary amines but at low rate [60]. Acrolein also offers slow reaction with guanyl functional group of guanosine and arginine [57]. Acrolein on reaction with primary amines generates Nε-3-formyl-3,4-dehydropiperidinyl adduct (FDP) [61]; whereas with nucleic acid and arginine it forms cyclic hydroxyl-N-propanyl adducts [23]. Potency of acrolein to crosslink proteins is same as HNE but more than MDA or hexanal [62]. Acrolein biological action is ascribed to its ability to modify thiols through depletion of GSH to evoke toxicity or via modification of cysteine active site like phosphatase to inhibit their action [23].

#### 2.3.4. Isolevuglandins (IsoLG)

The IsoLG are isoketals or 4-ketoaldehyde regio- and stereoisomers. These belong to the prostaglandin/isoprostane family of eicosanoids that are generated enzymatically via cyclooxygenase and non-enzymatically via LP. The IsoLG undergoes extremely fast reaction with primary amines, but slowly with nucleic acids or guanidinyl group such as arginine, and very poorly with thiols or imidazoles [63,64,65]. The IsoLG high reactivity against primary amine is based on initial imino adducts ability for successive reaction of 4-keto group with nitrogen to produce pyrrole adduct [66]. The pyrrole adduct easily get oxidized to produce lactam and hydroxy lactam adducts, and dimers/trimers of pyrrole [61,67]. The IsoLG have ability to crosslink proteins at very low concentrations. It exhibit inflammatory action by activating dendritic cells, endothelial cells and macrophages [68,69,70]. The dendritic cell take on IsoLG modified proteins and present neo-antigens to T cells which activate them, thereby results in hypertension and other vascular disorders [70,71]. The IsoLG reaction with protein modifies the ion channel and amyloid building protein, thereby develops arrhythmias [72,73,74]. Modification of IsoLG can block proteins degradation via proteasomal pathway and manifest in endoplasmic reticulum stress [68,74]. The antiinflammatory action of IsoLG via signaling pathway is yet unclear. However, receptors for advance glycation end products are known to assist in biological actions of IsoLG modified phosphatidyl ethanolamine [69].

#### 2.3.5. Hexanal

Hexanal, the monoaldehyde produced from ω-6 PUFAs oxidation, is nonreactive when compared with other RCS. In several diseases, the level of hexanal is a useful tool for the measurement of LP [75,76]. Hexanal reaction with primary amines (lysine) offers reversible imine adducts. Hexanal are very less reactive against thiol compounds (GSH or alpha-lipoic acid), and majority of imidazole compounds. As hexanal exhibits poor reaction against nucleophiles, thereby causes less development over hexanal scavengers [77].

#### 2.3.6. Malondialdehyde (MDA)

The compound 1,3-dicarbonyl malondialdehyde (MDA) is a highly prevalent RCS that is produced non-enzymatically from PUFA [78]; and enzymatically from prostaglandin (PGH_2_) via thromboxane synthetase that offers MDA and thromboxane (TXA_2_) in the ratio of 1:1. The MDA can be used as biomarker for LP as it reacts with thiobarbituric acid (TBA) to offer intense coloured chromogen called thiobarbituric acid reactive substances (TBARS) assay as forms TBA-RCS. The MDA reacts slowly as compare to other RCS. For example, it takes 1 h for 75% of HNE and 5 h for 50% of MDA (each at concentration of 100 μM) to react with 1 mM BSA [23]. The MDA exhibit insignificant reaction with thiols (GSH) at physiological pH. The MDA offers Schiff base on reaction with primary amines (lysine) that rearranges to alkenal adduct [79]. This further continues to react with other lysine residue of proteins to form protein crosslinks. Additionally, multiple MDA undergoes condensation reaction with primary amines and offer dihydropyridine. The MDA also react with nucleic acids (deoxyguanosine and -adenosine) to offer stable adduct like pyrimidopurinone (predominant) and adenine propenal that is highly reactive against lysine in comparison to MDA [80,81].

#### 2.3.7. Methylglyoxal (MGO)

This RCS is a dicarbonyl compound that is produced via glyoxlase based oxidation of fatty acids, carbohydrates and amino acids. In diabetes the MGO formation is highly prevalent attributed to high glucose level. The MGO exhibits slow reaction with α-amino group of amino acids; fast reaction with histidine, cysteine, or lysine; and very fast reaction with arginine [58]. Reaction of radiolabeled MGO with BSA presenting 50% of reaction after 6 h and completing reaction in 2 days, suggests that the rate of reaction of MGO with proteins is comparatively slow. During MGO reaction with guanidine (arginine) offers arg-pyrimidine, 5-methylimidazol-4-one (MI) and tetrahydro- pyrimidine (TP) [82]. The MGO and lysine reaction offers stable adduct of N-carboxy-ethyl-lysine (CEL) and imidazolium cross-links [83,84]. The MGO exhibits fast reaction against thiols (cysteine) to offer reversible hemi-thioacetal adducts. The amount of MI and CEL protein adducts are very high in numerous diseases like: atherosclerosis, renal disorder and diabetes (conditions in which glyoxalase expression is reduced) [85].

## 3. RCS Actions

### 3.1. Cytotoxic Effects of RCS

Under physiological condition, the RCS metabolic intermediates, like acetaldehyde, dioxyacetone phosphate and glyceraldehyde-3-phosphate are maintained at the low and steady-state concentration via rapid utilization by the subsequent steps in the relevant pathways. However, excessive RCS concentration (due to uncontrolled production of by-products during enzymatic reactions) may confer cytotoxicity. Additionally, biological effects exerted by the by-products are more stronger than their parental RCS [3]. The most common AGEs/ALEs include carboxymethyl phosphatidyl ethanolamine and carboxymethyl guanosine (derived from GO and MGO interaction with nucleic acids and phospholipids respectively), GO-lysine dimer, MGO-lysine dimer, carboxymethyl lysine, carboxymethyl cysteine, and arg-pyrimidine (derived from protein modification) [3]. These adducts are malfunctioned in nature as undergoes extensive covalent modification that led to a conformational change and distorted catalytic site. Further mechanistic investigation revealed that these adducts upregulate NF-κB signalling pathway via their interaction with receptor for advanced glycation end products (RAGE) to produce inflammatory cytokines (like IL-1, IL-6 and TNF-α) that are detrimental to cellular and organ dysfunction [86,87].

Interestingly, research work of Coughlan et al. revealed detection of oxidative stress (OS) in primary rat mesangial cells after exposure to AGEs or overexpressed RAGE in normoglycaemic conditions, along with significant accumulation of mitochondrial superoxide in hyperglycaemic conditions. This finding indicates that blocking ROS production could be a strategy to treat diabetic patients [88]. Proteins are the most studied target of RCS and AGEs/ALEs. Single-molecule approach revealed that proteasome usually target soluble oligomers assembled from ubiquitin-modified proteins and alter the aggregation process, by forming a structurally distinct aggregates to facilitate their excretion [89].

However, AGEs/ALEs are reported to further modify the proteins through continued covalent interaction [3], thereby inhibiting proteasomal degradation and leading to accumulation in cells or tissues. Carbonyl group get introduced in the proteins via oxidative pathways, either through direct interaction with ROS, or ROS firstly interact with molecules like lipids or sugars, to generate RCS. Protein carbonylation results in irreversible protein dysfunction, and its unwanted accumulation causes cell death that confers cellular or tissue injury (Figure 3a) [90,91]. High protein carbonyl level is detected at cellular and tissue levels in many chronic diseases, suggesting their potential role in disease pathogenesis (Figure 3a). For instance, study of Renke et al. demonstrated that plasma protein carbonyl quantum in children having juvenile RA was high as compare to healthy children (1.36 ± 0.68 vs. 0.807 ± 0.16 nmol/mg protein). This finding suggests their participation in inflammatory process and disease progression [92].

Additionally, it was discovered that the tracheal aspirates of premature infant with extremely low birth weight (less than 1500 g undergoing ventilation therapy) had a higher content of protein carbonyls (0.44 nmol/mg) in comparison to infants of birth weight more than 1500 g (0.29 nmol/mg protein). The high protein carbonyl concentration is accompanied with high myelo-peroxidase (MPO) activity as measured through oxidation of tetramethylbenzidine [95]. This finding reflects a possible contribution of neutrophil-derived RCS to chronic lung disease. The high protein carbonyls level was also detected in the severe form of pancreatitis, which is remarkably higher than the mild form (median 0.099 and 0.043 nmol/mg protein, respectively, *p* = 0.0016) [96]. This observation demonstrates the substantial evidence of protein oxidation measured by enhanced plasma concentration of protein carbonyls within 24 h of the onset of symptoms and persisted for five days of hospital admission, which was significantly higher in severe form of pancreatitis.

Apart from these, it is reported that RCS (malonaldehydes) and protein carbonyls are also implicated in causing cellular and tissue injury in ND particularly in mild cognitive impairment (which is an early condition of AD) [97]. Collectively, the cytotoxicity of RCS at cellular and tissue levels is explained in Figure 3a. High ROS levels can produce cytotoxic RCS, like: keto-aldehydes, α,β-unsaturated aldehydes and dialdehydes via the oxidative decomposition of PUFA and sugars. The accumulated RCS, particularly dicarbonyls interact with nucleophilic site of protein, nucleic acid, and aminophospholipid. The interaction may lead to irreversible modification and formation of adducts and cross-inks including AGEs/ALEs and carbonylated proteins. High expression levels of AGEs/ALEs, their receptors, and RAGE activate various signaling pathways including NF-κB. Subsequently, activated NF-κB stimulates production of inflammatory cytokines (IL-1, IL-6, and TNF-α), where their proinflammatory response could lead to cellular dysfunction. Meanwhile, protein carbonylation is associated with irreversible loss of function, causing accumulation in cells or tissues because specific proteosomes cannot degrade them. Such action manifests in cell or tissue injury that often leads to cell death [97].

### 3.2. Beneficial and Detrimental Effects of RCS (Receptor Level)

The RCS concentration determines its beneficial and detrimental roles in target cells or tissues. The total concentration of RCS derived from LP is reported to be less than 1 µM under physiological conditions [17]. Similarly, non-enzymatic LP derived RCS, such as 4-HNE and MGO, are detected in the range of 0.3–0.7 µM and 0.12–0.65 µM, respectively [10,23,98,99]. Several studies have discovered the beneficial role of RCS as an anticancer agent towards different cancer types, including prostate and colon cancer [98,99,100]. Furthermore, the studies revealed that RCS suppress cancer cells progression by decreasing the activation of 1,3-dioxo-1,2-dithiolane moiety of thiazole comprising 18-membered lactam ring of leinamycin (LNM) antibiotics; via activation of mitochondrial apoptosis pathway; or by inhibition of expression and activities of lipoxygenase (LOX), cytochrome P450 (CYP) and cyclooxygenase (COX) [100,101,102]. Beside this RCS also demonstrate antibacterial, antiprotozoal, antifungal and antiviral activities by disturbing the ROS homeostasis [103,104].

Another beneficial role of RCS is that they can regulate cellular signaling and/or transcription. It has been reported that 4-HNE at low concentration can induce stress-activated protein kinase/c-Jun N-terminal kinase (SAPK/JNK) signaling pathway (that is involved in immune response, stress adaptations and organogenesis during mammal’s development through regulation of various biological activities like cell survival, apoptosis and proliferation), either by direct binding or activation of redox-sensitive mitogen-activated protein kinase (MAPK) cascade [105,106]. The activated SAPK/JNK signaling is then translocated into the nucleus with subsequent JNK-dependent phosphorylation of c-Jun followed by binding of transcription factor activator protein (AP-1) (Figure 3b) [106]. Additionally, 4-HNE and MDA, which are lipid peroxidation-derived RCS are involved in stress adaptation, through the modulation of Nrf2/Keap1 signaling pathway via binding of Nrf2 with the electrophile response element/antioxidant response element (EpRE/ARE) [22,107]. Basically, Nrf2 is an antioxidant transcription factor that assist in maintenance of redox homeostasis at cellular level. The activity of nuclear factor erythroid 2-related factor Nrf2 (NF-E2-related factor 2) or transcription factor depends upon redox-sensitive inhibitor Kelch-like ECH-associated protein 1 (Keap1). In normal condition, transcription factor Nrf2 bounds to KEAP1 (that act as negative regulator by promoting Nrf2 ubiquitination for cytosolic proteosomal degradation).

In the case of oxidative attack, Keap1 structure undergoes a conformation change that make it unable to form a complex with Nrf2, resulting in a cytosolic accumulation of Nrf2 [3]. Once entered the nucleus, Nrf2 binds to EpRE/ARE to upregulate the cytoprotective antioxidant genes expression to encode for GSH reductase, quinone reductase, superoxide dismutase, catalase, peroxiredoxin, heme oxygenase, thio-redoxin reductase, GST, γ-glutamylcysteine synthase, GSH peroxidase, and other defensive proteins [107,108,109,110]. Nonetheless, recent studies highlighted that KEAP1 comprises multiple stress sensors and inactivation modalities, which together allow diverse cellular inputs (not limiting to Keap1/Nrf2 system) [111]. Accumulating evidence have demonstrated that an increase in the steady-state concentration via imbalance of RCS production and elimination could lead to several detrimental effects. One example is that an accelerated formation of irreversible modification of AGEs/ALEs could lead to rapid upregulation of RAGE expression (member of immunoglobulin superfamily) [112]. Consequently, such high expression further lead to ligand activation of key cell signaling pathways, like: p21ras, protein kinase C, MAPK, cdc42/rac and NF-kB [113,114]; thereby causes cellular dysfunction. For instance, the activation of the receptor of NF-kB can regulate expression of various genes that encode for ET-1, VEGF, TGF-β, TNF-α, IL-1 and IL-6 [115]. In addition, NF-κB also controls the transcription of genes related to cytokines, adhesion molecules and ROS/RCS generating enzymes (like NADPH oxidase and superoxide dismutase) [3]. Preclinical studies indicated involvement of AGEs/ALEs in progression and development of diabetic nephropathy [116,117], and hepatic inflammation [118]. Beside this, a study reported D-galactose-induced induction of neuronal insulin resistance by high cholesterol diet via interaction between AGEs and phosphoinositide 3-kinase (PI3K)/Akt signaling pathway [119]. The increase in dietary and extreme formation of endogenous AGEs stimulates OS and AGE/RAGE axis. This further increases the inflammation in the tissues, and induces insulin resistance, thereby enhances the progression of abdominal obesity [120,121].

Reports have shown the effects of ALE/AGE, particularly those contained in low-density lipoprotein (LDL) fraction of cholesterol in pathogenesis of atherosclerosis by promoting vascular tissue inflammation and remodelling in the initiation of hypertension through receptor and direct mechanism [8]. Moreover, it is demonstrated that presence of MGO (a potent inducer of AGE) favours cancer cell proliferation, migration and resistance to apoptosis due to “Warburg effect” (the increased ability of glucose uptake and fermentation to lactate, leading to induction of insulin, growth factors and inflammatory cytokines secretion) [122]. To support the implication of RCS in cancer progression Taguchi et al. also showed that RACE-amphoterin promoted tumour growth and metastases by activating p44/p42, p38 and SAPK/JNK MAPKs [123]. The association of RCS and its detrimental and beneficial effects are depicted in Figure 3b.

An elevated level of RCS may cause detrimental effects, particularly, induction of metabolic syndromes, such as: glycation, glucose intolerance, hyperglycaemia, abdominal obesity, hypertension, inflammation, and renal injury. Besides, RCS have been identified as an important glycation indicator in living microorganisms. High RCS levels can also elicit signaling pathways of various molecules via RAGE interaction (including NF-κB, p21ras, MAP kinases, PKC, and cdc42/rac). Among these NF-κB can regulate the gene transcription for several molecules (ET-1, VEGF, TGF-β, TNF-α, cytokines, adhesion molecules, and ROS/RCS generating enzymes). All the molecules regulated by these pathways can lead to cell property dysfunction. Furthermore, RAGE interact with amphoterin to implicate in tumour growth and metastases via activating of MAPKs. Contrarily, low levels of RCS are reported to be beneficial, particularly acting as a second messenger to activate several signaling pathways, for example SAPK/JNK and Nrf2/Keap1 to regulate cell proliferation and response to different stresses. Additionally, RCS has long been demonstrated to have anticancer activity by inhibiting metabolic pathways, such as: glycolysis and mitochondrial respiration [123].

## 4. RCS Management

### 4.1. Endogenous RCS Metabolizers

Some endogenous enzymes can metabolize (detoxify) the reactive carbonyl species like aldehydes that are highly toxic products of glycoxidation or lipid peroxidation. Such metabolic process involves phase I and Phase II reactions. The metabolic enzymes that commonly participates in such detoxification process includes aldo-keto reductase (AKR), cytochrome P450 (CYP450), glutathione-S-transferase (GST), alcohol dehydrogenase (ADH), aldehyde dehydrogenase (ALDH), and carbonyl reductase (CBR). Although, several metabolic studies have been performed over various carbonyl species, but still the exact mechanism for enzyme activity is unclear [124].

#### 4.1.1. ALDH

The human body comprises seven categories (ALDH1-7) for seventeen ALDH enzymes. Members of ALDH 1 and 3 oxidizes the lipid-derived aldehydes (like: *trans*-2-nonenal, *trans*-2-octenal, MDA and acrolein), ALDH2 metabolizes acetaldehyde, and ALDH5A metabolizes HNE [125]. Over expression of ALDH1A1 exhibit cells protection against *trans*-2-nonenal but there is no effect over lipid aldehydes. However, the ALDH3A1 enzyme have ability for complete blockade of HNE induced apoptosis and protection against these two aldehydes. This reveals ALDH3A1 ability to protect against aldehydes generated from LP. A study by Townsend et al. revealed the potential of ALDH3A1 to protect against HNE-induced apoptosis in epithelial cells of cornea [126]. In adipocytes the overexpressed ALDH3A2 exhibits protection against HNE, with no protection against MGO. Study reveals that individuals that are deficient of ALDH2 exhibit higher risk for AD and CVD. The PC12 cells that are ALDH2-deficient exhibit high sensitivity towards HNE and other oxidants. This supports the ability of ALDH2 to metabolize lipoxidation products and thereby safeguard against OS. A previous study reflected the involvement of ALDH2,3 and 5 enzymes in the aldehyde-derived pathway [127].

#### 4.1.2. CYP450

CYP450 participates in the metabolism of a wide range of products via C-H bond hydroxylation or epoxidation. The CYP subfamilies CYP4A and CYP3A have the ability to oxidize HNE (the LP product) [126]. 4-Hydroxy 2-nonenal (HNE) is reported for its implication in the tissue damage, aging associated injury, and other pathological conditions like cancer, inflammatory complications, diabetes, multiple sclerosis, AD and CVD [128,129], so oxidation of HNE by CYP subfamilies CYP4A and CYP3A highlights the importance of CYP450 enzyme as a therapeutic target for the treatment of various chronic diseases.

#### 4.1.3. Reductases

The enzyme family of aldo-keto reductase (AKR), alcohol dehydrogenase (ADH) reductase and short chain dehydrogenase reductase (SDR) are known to reduce the unsaturation in aldehydes or aldehyde to alcohols [9]. The HNE reduction by the sensitive enzyme of 4-methylpyrrazole revealed ADH’s involvement in aldehyde metabolism. The ADH belonging to medium chain dehydrogenase reductase (MDR) family plays an important role in oxidation of alcohols and aldehydes. For example ADH1 metabolizes *trans*-muconaldehyde, the benzene metabolite [130]. In human there are six members characterized in SDR family, namely: CBR1, CBR3, hydroxysteroid dehydrogenase (HSD), short chain retinol dehydrogenase (DHRS4), DHRS2, and xylulose reductase (DCXR). Among all these six enzymes families only CBR1 is involved in LP products, such as 4-oxonon-2-enal [131]. Study suggests AKR (belonging to NADPH-dependent enzymes family) role in the aldehyde reduction to alcohol. The AKR family is composed of 100 candidates, of which 10 are characterized in humans and falls under subfamilies of AKR1A-D and 7A. The AKR are reported for their involvement in metabolism of prostaglandins, steroids, sugars, and several aldehydes (generated due to OS). A study suggested AKR1A reduces LP and glycoxidation products (such as MGO, 3-deoxyglucosone, HNE and acrolein); AKR1B to reduce carbohydrates (glucose, MGO, and other triose) and HNE; ALKR1C to oxidize polycyclic aromatic hydrocarbon trans-dihydrodiol proximate carcinogen, and toxic aldehydes; and AKR7A to detoxify aldehydes, HNE, acrolein, and MGO [9].

#### 4.1.4. GST

GST is a dimeric protein that acts as a heterodimer for GSH conjugation. The conjugation of GSH is an important route for detoxification of OS induced aldehydes. The increased oxidant levels may deplete GSH level that compromises the GST conjugation. The GST comprises several families such as α-GSTA1-5, ω-GSTO1-2, µ-GSTM1-5, π-GSTP, θ-GSTT1, and ζ-GSTZ. The GST detoxifies several RCS (aldehydes), like: GSTP1-1 catalyzes GSH conjugation with short chain α,β-unsaturated aldehydes (acrolein and crotonaldehyde), GSTA4-4 catalyzes HNE, GSTM1-1 and GSTA1-1 catalyzes conjugating long chain 4-hydroxy-α,β,unsaturated aldehydes [132].

#### 4.1.5. Glyoxalase

The glyoxalase (I and II) enzyme metabolizes α-oxoaldehydes and methylglycol and forms α-hydroxyacids. This enzymatic system is GSH-dependent, such that in case of GSH depletion this system becomes inefficient [133].

### 4.2. Carbonyl Metabolizing Enzyme (CME) Inducers

The carbonyl detoxication may also be attained by increasing the CME activity via activation of enzymes or increasing levels of enzymes themselves, depending upon enhanced transcription, mRNA stabilization, or improved translation. Several studies highlighted range of natural and synthetic compounds that may enhance the expression of CME in different tissues, for example: sulphoraphane, 3*H*-1,2-dithiole-3-thione (D3T), benzyl isothiocyanate, phenethyl isothiocyanate, resveratrol, butylated hydroxyanisole, 3-methylcholanthrene, dithiolethiones, bezafibrate, β-naphthoflavone and other drugs [9]. The CME inducers benefit via two pathways which depends upon regulatory elements (promoter of induced genes). First the elemental pathway is xenobiotic response element (XRE), that relates to AH receptor (AhR) [134]. The second elemental pathway is antioxidant or electrophile response element (ARE/EpRE) that relates to Nrf2 the transcription factor [135]. The Nrf2 is endpoint of major pathway that controls range of antioxidant and protective enzymes. The Nrf2 availability depends upon redox sensitive regulatory protein Keap1 and it is a down-stream target of AhR. These two pathways are overlapping. Pathway of Nrf2/ARE represents adaptive response against OS, as induced enzymes are considered as protective and antioxidant [9].

#### 4.2.1. Inducers of ALDH

The induction of the ALDH group of enzymes may be enhanced using different therapeutic agents. For example: the level of ALDH3A (the HNE inhibitor) may be enhanced in human breast carcinoma cell lines by treatment with 3-methylcholanthrene via activation of XRE [136]; ALDH3A2 (the fatty ALDH) may be enhanced using bezafibrate via peroxisome proliferator activated receptor α-dependent mechanism; ALDH2 (the inhibitor of hypertension, myocardial infarction and AD) can be enhanced by sulphorophane; ALDH1A3 and ALDH2 may be enhanced by butylated hydroxyanisole (BHA) via ARE/Nrf2 mechanism [9]. Report suggest 3,4-Dihydroxyphenylacetic acid (DHPAA) as potential inducer of ALDH in murine hepatoma Hepa1c1c7 cells, thereby protects the cells from acetaldehyde-induced cytotoxicity [137].

#### 4.2.2. Inducers of AKR

The aldo-keto reductase enzymes like AKR1B are induced by MG and HNE via an OS-linked mechanism. In the case of cardiomyocytes aldose reductase is induced by D3T via the ARE/Nrf2 mechanism [138]. In vivo experiments on mice suggest AKR1B3 induction in reaction to D3T (Nrf2 dependent). Study highlights that AKR1A4 upregulation by D3T (as chemoprotective response); AKR7A1 (regulated by Nrf2 mechanism) in rat responds to chemical inducers (diet chemoprotectors and some drugs), and analysis of AKR7A1 gene promoter revealed presence of ARE elements. The AKR7A2 is elevated in AD which indicates its induction by OS. The AKR7A5 is shown to be induced by sulphorophane by a Nrf2-dependent mechanism which suggests its potential to enhance the metabolism of reactive aldehydes [9]. The AKR1C is known to be induced by chemopreventive agents, for example, AKR1C2 is enhanced by phase II inducers and AKR1C1 is induced by polycyclic hydrocarbons, electrophiles and OS. Drugs like sulforaphane, benzyl isothiocyanate and phenethyl isothiocyanate enhance the protein levels of AKR1C1 in human colon cell lines by 11-fold and 17-fold which correlated with protection against chemical stress. The microarray experiment in mouse exhibited AKR1C13 regulation by sulphorophane. Although, AKR1C enzymes are not efficient to remove HNE or MG, hence physiological relevance of their induction to reactive aldehyde load is still unclear [9]. Studies suggest the diuretic agent ethacrynic acid (EA) as AKR inducer, thereby act as potential approach in the treatment of hypertension [139].

#### 4.2.3. Inducers of CBR

Very few studies are conducted over upregulation of carbonyl reductase (CBR1) by chemo protective agents. The microarray experiment identified CBR1 to be responsive against range of inducers, including phenethyl isothiocyanate in mouse liver, D3T in mouse, and sulphorophane in mouse. Induction of CBR using these agents could be potential approach in the treatment of chronic disease such as down syndrome (DS) and Alzheimer’s disease (AD) [140]. The CBR1 expression depends upon transcription factor Nrf2 [9,141].

#### 4.2.4. Inducers of GST

Several GST (Nrf2/ARE-dependent enzymes) genes namely GSTA1-2, GSTM1-4 are induced by chemopreventive agents (like BHA, ethoxyquin, and oltipraz) and phytochemicals (like indole-3-carbinol, sulforaphane, and coumarin). A microarray study identified range of inducible GST that are Nrf2-dependent [9]. The CME like GSTA2, GSTM1-3, and GSTT1 are induced by phenethyl isothiocyanate (PEITC) in mouse liver via Nrf2-dependent mechanism [141]; the CME like GSTA2, GSTA4, GSTM1-3 subunits are induced by sulphorophane; the CME like GSTM1 and 3 are induced by BHA; and CME like GSTM1-4, GSTA2, and GSTT2 are regulated by D3T [9]. The most significant CMEs of these in terms of aldehyde detoxication are GSTA4, GSTM1, and GSTP1 that conjugates short and long chain a,β-unsaturated aldehydes. A study reports induction of endogenous GSH level and GST by D3T, which is connected with protection against acrolein- and HNE-induced toxicity [9]. Investigation reports clofibrate as inducer of GST, thereby acts as potential approach for the treatment of hypercholesterolemia [142].

#### 4.2.5. Inducer of Glyoxalase

The glyoxalase enzyme is reported to metabolize the α-oxoaldehydes and MGO to form α-hydroxyacids. [133]. Study reports sulforaphane to increase the expression and activity of glyoxalase 1 enzyme that degrades MGO (precursor for advanced glycation end products that are associated with diabetic complications and other age-related diseases). Therefore, this sulforaphane could be a potential approach for the treatment of diabetes and age-related complications [143].

### 4.3. RCS Scavengers

Scavengers are an alternative approach to RCS that can mitigate the dangerous effects of ROS without repealing of normal signaling that is mediated by ROS [2]. Scavengers use additional functional groups close to the primary nucleophiles, that catalyze the rate-limited reactions or assist to form irreversible and stable products [2]. Even in the case of a 1000-fold enhanced rate of reaction, the scavengers must be in μM concentration to compete with the endogenous nucleophiles available in tissue in mM concentration [63]. For this scavengers must possess adequate bioavailability and should act on tissues without toxicity [63,144]. Multiple classes of RCS scavengers are needed based on the fact that each RCS has different ability to modify specific functional groups. Depending upon the various functional groups the RCS scavengers may be utilized in the form of experimental probes to determine the involvement of a specific RCS type in a disease process [145].

#### 4.3.1. Thiol-Based Scavengers

Endogenous thiols like lipoic acid, GSH and N-acetylcysteine are utilized as dietary supplements, but high levels of these substances in the human body warrant the development of pharmaceutical thiols having high reactivity [8,146,147]. Investigations report that the reactions of amifostine and 2-mercaptoethanesulfonate (MESNA) with acrolein are faster than GSH. Research suggests a high use of MESNA and amifostine in human clinical conditions [148,149].

##### MESNA

The MESNA is a synthetic molecule that is widely applied as a systemic protective agent against chemotherapy toxicity, but it is primarily used to reduce hemorrhagic cystitis induced by cyclophosphamide that generate acroleins as a byproduct and thereby incites urotoxicity [150,151]. A recent study screening thiol compounds to scavenge acrolein (RCS) and reduce the urotoxicity in the rats treated with cyclophosphamide revealed MESNA as a highly potent scavenger [152,153]. In a traumatic brain injury (TBI) model the histopathological and biochemical analysis concluded that MESNA exhibits substantial neuroprotective effects over TBI by reducing LP and increasing antioxidant activity, whereas no antiapoptotic activity of MESNA could be shown. Moreover, MESNA exhibited less effect than methyl prednisolone (MP) in protection of brain tissue over TBI. Depending upon such promising results, MESNA is proposed to have future clinical therapeutic application in TBI [154].

##### Amifostine

Amifostine is a prodrug which is hydrolyzed by alkaline phosphatase into an active metabolite (WR1065). Acrolein reactivity with WR1065 is three times faster than MESNA [155,156]. Apart from scavenging of acrolein or related RCS, the WR1065 also causes scavenging of free radicals and induction of cell anoxia [157]. The scavenger amifostine is applied in conjunction with radiation therapy in case where mucositis and xerostomia limits the radiation dose [158]. A recent meta-analysis of 17 clinical studies covering 1167 patients revealed amifostine to reduce mucositis, xerostomia and dysphagia with relative risks of 0.72, 0.7 and 0.39, respectively, in patients undergoing radiotherapy [159]. Amifostine and WR1065 are claimed to be the best candidates for RCS scavenging in clinical conditions besides cancer, however only a few studies are reported. In a Parkinson’s disease rat model, WR1065 reduced RCS levels and protected against motor imbalance. Amifostin is also reported to reduce the reperfusion injury/ischemia in rabbits exposed to the temporary occlusion of the descendent thoracic aorta. Hopefully, the efficacy of amifostine will be further evaluated in such clinical conditions [160].

#### 4.3.2. Imidazole-Based Scavengers

Like thiols, imidazoles participate in Michael addition reactions to target α,β-unsaturated carbonyl compounds [161,162]. Unlike thiols, imidazoles do not undergo oxidation. Despite their low reaction rate against α,β-unsaturated carbonyl compounds, the highly stable imidazole containing scavengers possess efficacy like thiol scavengers. Carnosine is one of the most investigated imidazole-based scavengers [163], with high potential in the treatment of diabetes, cardiovascular and neurodegenerative diseases [164,165].

#### 4.3.3. Aminomethyl Phenols

The application of 2-aminomethyl phenol as a dicarbonyl scavenger was first discovered in studies utilizing pyridoxamine (PM) or vitamin B6 [166,167]. The wexcellent bioavailability and low toxicity of the primary amine PM justifies the rationality of testing its ability to scavenge MGO and other glucose-derived RCS in circulation [168]. The efficient scavenging of MGO in plasma by PM demands supplementation. This is because within cells PM is rapidly converted into pyridoxal phosphate (which is an enzyme co-factor lacking scavenging ability) [169,170]. The administration of PM provides protection against diabetes complications. In the beginning less consideration was given to the structural relationship of PM and its RCS reactivity in comparison to other bioavailable primary amines [171]. Such aspect was appreciated based on studies intended to examine the PM efficacy against IsoLG. The high reactivity of 4-ketoaldehydes convinced investigators to recognize that finding an effective scavenger against this RCS type will be difficult. This is because it requires a compound with similar reactivity but at much higher biological concentration than lysine, or a similar biological concentration but with much faster reactivity for 4-ketoaldehydes compared to lysine. Depending upon first possibility numerous biological small molecules of primary amines including aminoguanidine, PM and glycine were screened to stop the IsoLG modification of radiolabeled lysine. Surprisingly, equimolar PM totally inhibited the lysine and IsoLG reaction, whereas equimolar aminoguanidine or glycine had insignificant effects [171,172,173].

##### Pyridoxamine (Pyridorin)

The PM is a hopeful drug of choice for the treatment of chronic diseases or disorders that are connected with RCS- and OS-assisted pathogenicity, such as diabetes and its complications including CVD, retinopathy and renal disease. Such properties of PM are based on its ability to inhibit the formation of advanced glycation and lipoxidation products by scavenging of RCS. PM might therefore prevent protein damage from lipid hydroperoxide-derived aldehydes such as ONE and HNE by trapping them. PM may prevent the lipid hydroperoxide-derived damage to proteins through trapping of ONE [167,174]. An in vivo study involving 28 weeks administration of PM to diabetic rats (induced with STZ) revealed PM’s potential to significantly inhibit renal disease based on a paramount reduction in the level of plasma creatinine and albuminuria [175]. The investigation reported that 12 weeks of administration of PM to KK-Ay/Ta mice markedly protected the kidney function based on the low albumin and creatinine ratio in urine. PM further exhibited its potential in reducing the insulin level (fasting), nitrotyrosine, renal N-carboxymethyl lysine (CML), and TGFβ1 levels. The PM treatment of db/db mice reduced kidney hypertrophy, proteinuria, podocyte loss, mesangial expansion and also the development of glucose intolerance. In an acute kidney injury ischemia/reperfusion model the pretreatment with PM before nephrectomy caused a reduction in fibrosis and injury score. The PM pretreatment also reduced serum creatinine levels after 28 days post-nephrectomy [176].

##### 2-Hydroxybenzylamine (HOBA) and 5′-O-pentyl-pyridoxamine (PPM)

The other 2-aminomethyl phenols that have undergone in vivo testing are HOBA (salicyl amine) and to a lesser extent PPM [2]. Based on their high lipophilicity and cellular scavenging in comparison to PM, HOBA and PPM are expected to have higher efficacy in comparison to PM [177]. The in vivo pharmacokinetic experiments with HOBA in mice revealed good oral bioavailability (38%) and partitioning through plasma in a majority of tissues that include liver and brain [178,179]. HOBA protects against cellular damage by scavenging the RCS. HOBA may help to protect against age-related pathologies when used prophylactically, but it cannot reverse pre-existing damage [180,181].

### 4.4. Natural RCS Scavangers

Evidence suggests that a high consumption of fruits and vegetables reduces cardiovascular and all other cause-based mortality [168]. However, in case of chronic diseases antioxidant supplements and multivitamins exhibited no benefits [182], but rather produced harmful effects [183]. Based on the fact of the potential of natural products to sequester RCS, the RCS-sequestering properties of natural products can be used as an effective preventive strategy in the treatment of chronic diseases [1].

#### 4.4.1. Carnosine (Endogenous)

l-Carnosine is a endogenous dipeptide (β-alanyl-l-histidine) that is most prevalent (~10 mM) in muscles and brain [184]. *C*arnosine, the natural dipeptide also has the potential to scavenge intracellular RCS and form unreactive covalent adducts [185], which are then excreted in urine [145], thus providing an additional mechanism of RCS detoxification. Studies report natural carnosine to be reduced in obesity, leading to a greater susceptibility to RCS accumulation [185,186]. The carnosine level is low in diabetes and AD patients, and modestly low in vegetarians, females and the elderly population. Apart from scavenging, carnosines act as a physiologic pH buffer, a hydroxyl radical scavenger, a redox-active metal chelator, a stimulator of nitric oxide synthesis and an activator of carbonic anhydrase [164,165]. Carnosine supplementation can lower the RCS level in rodent models, however, very few studies were conducted in humans [185]. There was a report on a comprehensive analysis of the quenching potential of carnosine and its analogues towards MGO and malondialdeyde as typical examples of reactive dicarbonyls [187,188]. Carnosine possesses the ability to selectively quench α,β-unsaturated carbonyl species [189,190]. However, the reported data reveals that carnosine and its derivatives are also reasonably active as quenchers of malondialdehyde, but that they are substantially devoid of reactivity against dicarbonyls [191].

Anserine (β-alanyl-l-methylhistidine) and carnosine are the two dipeptides that detoxify HNE by generating non-reactive adducts [192]. Histidine, the highly reactive nucleophilic protein residue is considered the key reactive site for HNE adducts [193]. Supplements of histidine dipeptides (HDs) are claimed to lessen the progression of hypertension, dyslipidemia and renal injury by lowering the extent of protein carbonylation and glycation [194]. Additionally, HDs are useful in various systemic oxidative and glycative stress [195,196]. Evidence suggest HDs can exert health-promotion effects by decreasing the AGEs/ALEs levels thus preventing damage of AGEs/ALEs-RAGE. Depending upon an individual’s genetic background, the genes-nutrients interaction may lead to differences in supplement bio-efficacy. Reports suggest such an interaction with vitamin E-haptoglobin and vitamin C-GST [197,198]. Studies suggest a connection between low serum levels of carnosine with diabetic nephropathy, such that CNDP1 (a carnosinase encoding gene) is connected with late onset of complications in diabetic patients. Patients with 5-6, 5-7, 6-6, and 6-7 alleles of CNDP1 gene exhibit high serum carnosinase action, but individuals with 5-5 allele that account for 1/3 of the population, were noticed to be less susceptible towards renal complications [199]. The high carnosinase expression increases carnosine degradation that leads to a lesser degree of renal protection by carnosine [1]. However, such a theory must be confirmed in human trials, as genes-nutrients interactions are an area that must be investigated for in depth understanding of natural products bio-efficacy.

#### 4.4.2. Plant Products

A study reports that black rice (containing anthocyanin, GABA, α-tocopherol, α-tocotrienols and γ-oryzanols) exhibit anti-hyperinsulinemic and anti-hyperlipidemic activity in *ob/ob* mice [200]. When expanding results from animals to humans, the determination of such activity in natural products offers a targeted preventive approach against chronic diseases. Additionally, using the mass spectrometric technique the effects of green coffee bean extract and *Vitis vinifera* (procyanidins) on carbonylation of proteins were demonstrated. The two extracts exhibited significant in vitro inhibition of HNE-induced ubiquitin carbonylation in a dose-dependent fashion [201]. Natural products screening for RCS quenchers is the first step for determination of potential candidates for target strategies to prevent OS-related chronic diseases. However, further understanding of gene-nutrient interactions and enhancement of the limited bioavailability/bio-efficacy of natural products are needed. Figure 4 summarizes various approaches to manage RCS.

## 5. Nanotechnology to Enhance the Bioavailability and Bio-efficacy of RCS Scavengers

Their limited bioavailability and bio-efficacy are the two major obstacles for RCS scavengers. This hinderance is attributed to their lack of solubility, instability and absorption. Administering megadoses is not ideal, but the development of effective delivery systems to enhance the bioavailability and bioefficacy offers hope for the development of ideal RCS scavengers [1]. Quite a large number of studies report applications of nanotechnology to improve the bioavailability and bioefficacy of nutraceuticals [202,203]. Nanodelivery systems like nanoemulsions, nanoliposomes, and nanomicelles that are biodegradable and biocompatible are reported to enhance the bioavailability of phytochemicals [204]. Application of epigallocatechin gallate in the form of nanoliposomes exhibited high stability and enhanced antioxidant potential [205,206]. Studies have suggested that the bioavailability of quercetin (a hydrophobic flavonol) can be improved by applying it in the form of nanomicelles [207,208]. Administration of quercetin in encapsulated form is reported to delay its metabolism and thereby maintain a high free quercetin level in plasma and target tissues [209]. The application of nanotechnology is also known to improve the bioavailability and bioefficacy of orally administered hydrophobic curcumin [210]. Recent advances in nanomaterial technology exhibit its high potential to enhance the bioavailability and bioefficacy of natural products [211].

The latest graphene discovery stimulated research on target delivery of bioactive compounds. Graphene is single thick layer of sp^2^-hybridized carbon atoms arranged in a honeycomb-like two dimensional (2D) crystal lattice structure [212]. Graphene’s unique atomic structure offers outstanding properties like a large surface area, biocompatibility, fast mobility, and super electrical conductivity [213]. The properties of graphene make it a most suitable material that possesses a large number of applications including quantum mechanics and biomaterials engineering such as new drug delivery nanocarriers, new generation biosensors and biological imaging probes [214,215]. In the recent two decades among nanomaterials for various drug delivery system, graphene, graphene oxide (GO) and graphene quantum dots have emerged as new competitive nanocarriers for drugs and natural products’ delivery.

In regards to treatments for spinal cord injury (SCI) and alcoholic liver disease (ALD) attributed to acrolein, improvement of the low oral bioavailability of hydralazine through traditional formulation is important. This is attributed to evolution of nanotechnology that allows local drug delivery directly to damaged cords. Successful development of polyethylene glycol-functionalized (PGF) mesoporous silica nanoparticles (MSNs) as a novel hydralazine delivery system by Cho et al. worked well during in vitro PC12 cell bioassay for 5 days in a non-linear fashion. The study supported the effectiveness of hydralazine in SCI and ALD attributed to the ability to rescue neurons without inducing toxicity. The PGF-MSN loaded with hydralazine offered significant neuroprotection to cells damaged by acrolein [216]. In another ex vivo study to scavenge acrolein, Cho et al. used PGF-MSN after crushing spinal cord injury and demonstrated the capability to restore action potential to near pre-crushing levels. The study revealed that recovery could be attributed to the PEG coating outside the nanoparticles [217]. Finally, to investigate scavenging of acrolein, Cho et al. used hydralazine encapsulated in chitosan nanoparticles with a diameter of 350 nm. Their results revealed the low encapsulating efficiency of chitosan nanoparticles that were able to reduce the death of acrolein-exposed mouse [218]. A study involved utilization of peptide amphiphiles (PA) that self-assembled in vivo in supramolecular nano fibers in a mouse model of SCI. Administration of PA after SCI reduced astrogliosis and cell death, however, it raised the number of oligodendroglia at the injury site. Treated mice exhibited improved motor capabilities and decreased apoptosis in and around the injury site [218]. Remarkably, studies suggest the limitations of bioavailability and bio-efficacy of scavengers can be improved by using selenium, melanin, and cerium oxide-nanoparticles that are known to boost OS defense systems in vitro and in vivo [219].

## 6. Detection of RCS

Studies reveal that in lower concentrations RCS exhibit beneficial effects in the human body, whereas in large concentrations RCS exert detrimental effects. Hence, there is an utmost need to investigate the concentration of RCS in the human body [3,10]. To understand the formation and biological importance of RCS, the availability of reliable and sensitive methods for detection (identification and quantification) of RCS in biological matrices is really challenging and a must. This is attributed to the physicochemical properties (high reactivity and low molecular weight), lower concentration in biological samples, higher hydrophilicity (unsuitability for reversed phase LC), and unsuitability of most detectors for direct detection of RCS. Most RCS exhibit limited absorbance attributed to an absence of chromophores and ionizable groups in their structures that cause poor RCS ionization yields and detection by electrospray ionization mass spectrometry (ESIMS) [220]. Such restrictions of analytical techniques for RCS profiling in biomatrices (serum, urine and cells) creates the need for RCS derivatization. Derivatizing agents (DA) for RCS contain functional groups that exhibit quick reactions with carbonyl groups and include a moiety to facilitate their detection, like 2,4-dinitrophenyhydrazine (DNPH), 7-(diethylamino)coumarin-3-carbohydrazide (DACHH), toluenesulfonyl hydralazine (TSH), dansyl hydrazine (DH) and dimethyl aminosulphonyl)-7-hydrazino-2,1,3-benzoxadiazole (DBD-H) [220].

The DA is selected based on the detector(s) available. Few derivatization agents help detect bicarbonyl group-containing protein-bounded lipids that still have one free carbonyl functional group. Initially RCS analysis was based on DNPH (the DA) and related reaction products (like Schiff bases). Such RCS analysis involved UV detection or UV detection coupled to HPLC intended to identify each individual aldehyde [221,222]. Such analytical methods were suitable for analysis of free RCS in vitro, but unsuitable for in vivo samples, which was attributed to a lack of sensitivity and selectivity. Later, for free aldehyde and ketone derivatives of DNPH the LC-MS and GC-MS analysis techniques were applied [223]. Currently, the use of DNPH is limited due to its limited solubility in aqueous solvent, explosive nature and deposition in the ion source of MS instruments [224]. Recently, several advanced methods were used for RCS profiling and quantification in bio-matrices. These analytical methods are categorized in two major categories, namely targeted analysis (for known RCS) and untargeted analysis (for unknown RCS).

### 6.1. Untargeted Analysis

Recently, several comprehensive oxolipidomic (OL) methods were suggested for identification of unknown RCS in different pathophysiological states, and thorough determination of oxidized lipids/related dissociated products in biomatrices. The first OL method applied carnosine as DA for identification of 4-hydroxyalkenals (like HNE) in oxidized lipids, followed by infusion over an electron spin ionizer (ESI) attached to a triple quadrupole mass analyzer [225]. The high reactivity of carnosine with alkenals and easy ionizability of MA assists in simple detection using ESI. The collision-induced dissociation (CID) of carnosine adducts cause fragment ions or neutral loss that assist in identification of unknown adducts. The hydroxyalkenal species structures were characterized based on their PUFA structures and molecular weight. The OL method when applied over lipid extract from mouse tissue identified four types of hydroxyalkenals such as: 4-HNE, 4-hydroxynondinenal, 4-hydroxydodecatrienal and 4-hydroxyhexenal. Quantitative determination was also done using a deuterated standard.

There is second more extensive OL approach that employs RCS derivatization by carbohydrazide (CHH) followed by ESIMS [226]. Detection of derivatized compounds and determination of dissociation sites and oxidative modifications was based on the specific fragment ions or neutral losses. This method identified in vitro oxidation products of free fatty acids (with signals at *m/z* 69) and phosphatidylcholine vesicles (with a signal at *m/z* 122), thereby determining the complexity of generated RCS. The CHH method was used to characterize the lipid extract of primary cardiac myocytes of rats that were pre-treated with 3-morpholinosyl- nonimine-*N*-ethylcarbamide (SIN1), a peroxynitrite donor [227].

There is third another OL method that analyzes oxidized lipids using TSH as the DA [224]. In terms of reactivity TSH is similar to DNPH, however it possesses higher volatility and improved solubility. The TSH acts as a universal DA for detection of RCS when coupled with sequential window acquisition of all theoretical (SWATH) fragment-ion MS at higher resolution. The TSH-SWATH method was utilized for thorough identification and quantification of known and uncharacterized RCS in biomatrices based on detection of characteristic fragment ions that were originated from DA substructures.

There is a fourth LC-ESI-MS method that applied DH as DA and uses a MS analyzer in selective reaction mode (SRM). This method monitors ions at *m/z* 236.1 ascribed to CID of derivatized RCS, whose [M+H]^+^ was scanned between 275 to 949 [228]. A fifth derivatization-free method is based on UHPLC-HRMS to identify the RCS. This approach is intended for analysis of RCS from 6 to 16 carbons and when using ESI in positive-ion mode exhibits signals for [M+NH_4_]^+^ and [M+H]^+^, whereas in negative-ion mode it exhibits signals for [M+HCOO^–^] [229].

### 6.2. Targeted Analysis

Recently, several methods were suggested for the quantitative analysis of known free RCS in biomatrices, such as gas chromatographic mass spectrometry (GCMS), LC–MS/MS, micellar-electrokinetic chromatography (MEKC) and several other derivatizing approaches. Analytical methods that target quite a large number of RCS, when analyzing RCS derived from lipid and glucose sources are highly appreciated. This is because in a pathological state there is cross-communication between protein modification by the two classes of RCS. An HPLC-coupled fluorescence method that applied 2,2′-furyl as DA for RCS detection was validated for quantitative analysis of MDA, ACR, GO and HNE in human sera with a detection limit ranging from 0.03–0.11 nmol/mL [230]. The validation study revealed that MDA was the most abundant RCS (10 nmol/mL serum concentration) in healthy individuals, followed by other RCS (aldehydes at 1 nmol/mL concentration). This method analyzed remarkably high serum concentration of HNE, ACR and MDA in diabetic and RA patients in comparison to healthy individuals. Another method involved derivatization of carbonyl groups using DBD-H to produce peroxyoxalate chemiluminescence [231]. This method was used to analyze MGO, ACR, CA and *trans*-2-hexenal concentrations ranging from 4.4 to 6.5 nM.

Despite the development of comprehensive OL methods in past years, several limitations and questions about RCS analysis in biological matrices are not yet answered. RCS may possess different stability, for example unsaturated aldehydes exhibit less stability in comparison to saturated ones, and this stability may affect the percentage recovery yield and accuracy of the RCS detection methods [220]. Thus, standardization and reproducibility of RCS recovery from bio-samples becomes important. This can be achieved through preparation of suitable sample by adding appropriate DA and spiking deuterated internal standards. However it is also essential to detect the RCS source using derivatization methods, to determine the RCS proportion in free or protein-bound form, and to determine the RCS proportion that may be produced from reversible protein adducts (like Schiff bases or MA). Additionally, the target analysis based on DA generally misses Schiff base protein-lipid adducts.

With the advancement of analytical methods, the comprehensive OL approach has identified more than 400 different RCS species [228]. This along with knowledge of other lipidic reactive species (nitrated fatty acids) that do not contain carbonyl groups emphasizes the structural assortment of reactive species that may modify proteins [232]. Considering that the consequences of modification depend upon the structures of adducts, a large number of distinct protein species may arise from lipoxidation. It is important to ascertain the quantity of free RCS and protein-bound RCS, since it assists model system studies in determining the function of lipoxidation using concentrations in the range of those happening in vivo. Hence, a combination of approaches is needed to realize the function of lipoxidation in pathophysiology.

### 6.3. Mass Spectrometric (MS) Approaches

#### 6.3.1. Label-Free MS

MS-based analytical methods are extensively used in the study of biosystems and protein-lipid adducts. MS analysis provides meticulous identification and quantification of compounds, with high precision, sensitivity and high throughput. MS provides molecular structural information to characterize the protein-lipid adducts both in vitro and in vivo. To characterize the adducts under pathophysiological conditions, the initial characterization of potential adducts can be done through analysis of peptides/protein-lipid adduct formulated in vitro using pure preparations of test peptides/proteins and lipid electrophiles. Lipid electrophiles owing to their high reactivity immediately forms adducts with -NH2 or -SH functional groups. Newer peptide-lipid adducts are easily detected through MS coupled with liquid chromatography (LC-MS) or electrospray (ESI-MS) or matrix assisted laser desorption/ionization (MALDI-MS). For protein-lipid adducts the MS characterization can be done using bottom-up or top-down proteomic approaches. The widely used bottom-up approach involves protein-lipid adduct enzyme hydrolysis (commonly with trypsin followed by tryptic peptides and peptide-lipid adducts) followed by LC-MS or MS/MS and MALDI-MS analysis [233].

In MS, the peptides-lipids adducts are detected as single protonated molecular ions [M+H]^+^ or multiple protonated molecular ions [M+nH]^n+^. However, the signals of structurally modified peptides are recognized in MS spectra based on the shift in mass of peptides that are structurally non-modified. For MA, the shift in mass due to structural change corresponds to the RCS molecular weight. However, for Schiff bases the mass shift corresponds to M – 18, where M is the RCS molecular weight and 18 amu is due to the loss of H_2_O during the adduct formation. Likewise, in the case of MDA-peptide adducts, the mass shift is +72 for MA and +54 for Schiff adducts having a single charged ion; and +36 for MA and 27 for Schiff adducts having doubly charged ions, as per the LC-MS study of MDA-β-lactoglobulin adducts [50,234]. For peptide-lipid adducts, the nature of oxidative modification and its location in the peptide backbone is generally confirmed using tandem (MS/MS) mass spectrometry.

The MS/MS technique generally involves CID, and information about peptide modification sites and adduct type is acquired by deriving typical y or b-type fragment ions or the existence of modified immoniun ions [235]. CID-MS/MS spectra exhibit characteristic carbonyl moiety losses, which form unmodified b- and y-ions, that may inhibit identification of the adduct position. This characteristic fragmentation pathway of loss of lipid electrophiles and modified immonium ions assists in defining the target reporter ion-based MS analysis, such as neutral loss scanning (NLS) or precursor ions scanning (PIS). NLS is used extensively to detect HNE-peptide adducts. MS^3^-NLS with 52 (triple charge), 78 (double charge) or 156 (single charge) methodologies are used to locate HNE-peptide MA [236,237], for example HNE-β-spectrin or HNE-cytochrome C oxidase adducts in human plasma [237,238]. Recent studies report the use of methods which do not cause labile bond fragmentation, for example: electron capture dissociation (ECD) and electron transfer dissociation (ETD) [239,240]. The MS/MS data acquired using ETD or ECD is highly informative as it exhibits abundant c- and z-type fragment ions of modified peptides. This is because carbonyl-peptide bonds exhibit more stability, preventing carbonyl moiety loss. Moreover, ETD is also involved in top-down approaches, as it assists in bigger peptides and intact protein analysis. Studies suggest protein and peptide adduct analysis with tiny lipid electrophile. For example under both in vitro and patho- physiological conditions, the protein adducts of these tiny lipids and α,β-unsaturated aldehydes (ACR, HNE and MDA) [233,241,242] are commonly detected and characterized [243,244]. Facts suggest cyPG protein adduct formation in various models of pathophysiological conditions using different techniques. Many studies have connected identification of modified proteins using metabolic labeling and cyPG-peptide adducts with MS characterization in positive mode by incubating peptides or proteins in vitro with cyPG [242,245].

The MS detection of endogenously formed prostaglandin-protein adducts is yet to be proven, and protein/peptide modification through lipid electrophiles when esterified to phospholipids is rarely studied [242]. Studies have identified phosphatidylcholine-generated RCS adducts with Cys, Lys and His in the synthetic peptides ApoA and B 100 [246,247]. This needs further investigation as protein-oxidized phospholipid adducts detected in HDL and human platelets comprise electrophilic oxidized phosphatidylcholine [248]. The number of protein-lipid adducts generated in regulated chemical conditions is quite low, so related spectra are easy to analyze. Information acquired with such methods helps identify adducts that would be generated for each lipid electrophile. Such methods benefit to determine the reactivity of specific lipid electrophile species. Also, the fragmentation pattern of lipoxidation adducts under MS/MS conditions can be identified, which will be useful in target analysis of bio-samples.

Analysis of complicated cell, tissue and biofluid samples requires many complicated, specific and targeted methods for identification and characterization of protein adducts. These require enrichment methods, chemical labeling and a bottom-up proteomic approach that combines enzyme reactions, chromatography and analysis using MALDI-MS, LC-MS and MS/MS. Methods to enrich and derivatize, followed by MS analysis improves the sensitivity and selectivity, which assist in identification and may be commenced after or before protein digestion. The aldehydes which react with lysine and arginine offer more problems as such modification might interfere with digestion of trypsin, that generates longer peptides which are difficult to sequence. The false identification of HNE adducts may happen as they might add the same mass as arginine. Usually, prior to enzyme digestion the lipid adducts are stabilized with NaBH_4_, followed by MS analysis [249]. The NLS scanning-based target analysis of derivatized or non-derivatized samples is a useful analytical approach to determine specific protein modifications due to particular lipoxidation products.

Report suggests application of combined MS approaches (MALDI-TOF and LC-ESI-MS/MS) to determine protein lipid adducts in bio-samples. Bottom-up methodology using HR-LC-MS/MS and target analysis under multiple-reaction monitoring (MRM) identified HNE-metalloproteinase rhMMP-13 adducts based on specific neutral loss of HNE in vitro and in chondrocytes of osteoarthritis patients [250]. The proteomic study involving coupling of MALDI-TOF/TOF coupled by SDS-PAGE and enzymatic digestion revealed that HNE targets liver fatty acid-binding protein (L-FABP) for modification [251]. In the liver steatosis model the RCS like HHE, HNE and ONE exhibited modification of nuclear proteins such as: chromodomain-helicase-DNA-binding protein 4 (chDBP), actin, heterogeneous nuclear ribonucleoprotein L (hnRNPs), and neuroblastoma differentiation-associated protein AHNAK using SDS-PAGE and LC-MS [252].

#### 6.3.2. Label-Based MS

Generally, protein oxidation is measured based on protein carbonylation or protein carbonyl groups formation. There are several simple, economical, and robust methods available to detect protein carbonyl groups [253]. There are some bifunctional lipidoxidized products that do not undergo reactions with proteins to form free carbonyls, but the direct oxidative deamination of the side chains of arginine, proline, lysine, and the other residues can generate protein carbonyls. Although, several studies support protein carbonyl formation in any disease, this may not be sufficient evidence for lipid oxidation until lipid oxidized products are tested with antibodies or modification of adducts is detected by MS. The selectivity and sensitivity of carbonylated protein detection can be improved through carbonylated lipid-protein adduct labeling before MS analysis [254]. The derivatization approach commonly exploits the reactivity of the free carbonyl groups of lipoxidized MA, that are highly reactive with free amino group-containing derivatizing agents. This method is not able to derivatize lipid-protein Schiff base adducts unless there is dialdehyde lipid electrophile. The DNPH is the most widely used derivatizing agent, because DNPH-labeled carbonyl peptides exhibit promising ionization and detection sensitivity. However, during RCS analysis the DNPH labelling offers a lack of selectivity as DNPH is also reactive against sulfenic acids [255].

Although DACHH is also reported for derivatization of lipid-bound carbonyls [256], other chemical labeling can be done [257]. The use of biotin-based hydrazide functional agents is another common chemical labeling method. Such a method enriches biotin-linked adduct through avidin chromatography and reduces unmodified peptides in MS analysis [239]. Therefore, two approaches can be adopted, firstly to enrich biotinylated proteins followed by enzyme digestion and MS analysis. Secondly, protein digestion followed by separation of biotinylated peptides using affinity chromatography [233]. There is increasing research on MS analysis coupled with enrichment methods using biotin for recognition of site-specific proteins modifications in complex proteomes in biosamples. A recent study identified a HNE-peptidylprolyl-*cis/trans* isomerase A1 (Pin1) adduct that was linked with catalytic cysteine (Cys113) by conjugating biotin with Pin1 tailed by LC–MS/MS analysis using click chemistry [258]. The treatment with biotin hydrazide (BH) followed by avidin pull-down LC–MS/MS or MALDI-TOF/TOF, assisted in recognition of in vitro modification of NAD-dependent deacetylase sirtuin-3 (zrSIRT3) the mitochondrial protein using HNE at Cys280 [259]. Additional investigation is required to determine the lipoxidation impact in health and diseases. In the future investigations must focus on new methods for in vivo detection of carbonylated proteins and lipid adducts.

### 6.4. Non-MS Approaches

Some metabolic labeling approaches that involve incubation of biosamples (cell cultures) with labelled analogues of oxidized species or RCS or their predecessors may be used in MS and non-MS methods. For example incubation of cell cultures with radioactive arachidonic acid (the precursor for prostaglandins or other lipid electrophiles) helped in the detection of incorporation of radioactivity into proteins [260]. As such methods do not affect the lipoxidized species structures, it is possible to preserve metabolism and interactions, but since this labeling approach involves radioactive material, imaging or enrichment procedures are not permitted. To study protein lipoxidation, the biotinylated analogues of lipid electrophiles species (example: cyPG) or their precursors are the commonly used probes. The benefit of several cyPG in tissue injury and inflammation has stimulated research to identify the modified proteins and novel therapeutic targets prior to interpretation of their actual pathophysiological importance. Protein modification using biotinylated cyPG has stimulated research on lipid species with similar structure or reactivity including isoprostanes and nitrated fatty acids. Several cyPG targets including NF-κB and transient receptor potential (TRP) channels were identified [261,262]. The cyPG were widely used owing to their stability (like stability of resultant adducts with proteins), and their susceptibility for derivatization via modification of carbonyl groups through various functional groups.

After the first report on biotinylated PGA2 and 15d-PGJ2, several other biotinylated analogues of arachidonic acid-derived lipid electrophiles were generated [260,263]. Such analogues help in exploring high affinity avidin-biotin communication intended to detect and enrich biosamples to purify and identify. However, this approach has the demerit that biotin groups impose structure restrictions that might disturb the interactions with proteins. Therefore, despite the fact biotinylated analogues can mimic several effects of their parent compounds, there exist important functional differences [263,264].

Additionally, none of the metabolic labeling methods so far is found suitable in humans. Fluorescent labels are applied in lipid electrophiles species in gels or cells to detect and quantify lipoxidized proteins [265,266]. These approaches when coupled with proteomic methods become suitable to identify lipoxidation potential targets. However, due to the bulky nature of fluorescent functional groups in the case of biotinylated labels, the validation of identified targets is compulsory and also biological interactions may occur. Facts suggest that click chemistry helps in monitoring the providance of HNE and oxidized phospholipids [258,267], and offers smaller labelling groups that generally do not hamper the interactions or metabolism of labeled lipids. Such a labeling technique helps in tag derivatization using alkyne-azide reactions ex vivo in tissue extracts, partially purified biosamples or pervious cells; thereby, facilitates imaging detection, enrichment etc. For example, this approach assists in HNE-induced cross-linking of Pin1 [258].

Lipoxidation acts as a selective process as it affects cellular proteins selectively that are guided through the structures of proteins and reactive lipid species [245,268]. The existence of intra-cellular redox status and small antioxidant molecules (example: GSH) is claimed as an imperative determinant for selectivity [268,269]. The lipoxidation can offer selective compartmentalization based on location of lipid electrophiles species formation or distribution of small molecules (example: thiols that may act as decoy) or the availability of oxidized lipid detoxification enzymes (example: GSTs) [265,270]. Hence, lipoxidation choreography could be imperative in its functional consequences. Thus, to detect lipoxidation selectively in specific subcellular environments, a few probes were synthesized which combine derivatization of lipid electrophiles (with biotin/fluorescent tag) and organelle-specific tags or target groups [271].

Generally, all such label-based techniques are applicable in both positive (direct) and negative (indirect) methods for detection of protein modification. Positive methods involve identification of modified targets, whereas in negative methods firstly protein modification is induced (using lipid electrophiles or oxidants) later signal reduction is observed using a labelled probe or vice versa [266,272]. Such methods could be used in vitro or in cell or tissue studies. These methods assist in studying the competition between parent and tagged lipids or interactions of lipoxidation and simultaneous modifications (oxidative or through endogenous or exogenous electrophiles like drugs) that upset the cysteine residue [273,274]. In the case of living cell studies these methods must be combined with some other methods as modifications that occur in cells exposed to oxidized lipids could be different and may involve lipoxidation along with various RCS oxidations related to RCS associated OS [273]. As per the earlier discussion over ex vivo lipoxidized protein derivatization using various reagents in addition to MS methods, the detection of derivatized compounds can be done using fluorescence spectroscopy, avidin–biotin methods or specific antibodies. However, a few derivatization techniques offer some limitations also. Lipoxidation may escalate the volume of carbonyl groups over proteins attributed to incorporation of a few lipid compounds that contain these groups.

The carbonyl moieties generated by oxidative modification of residues can be detected through techniques that involve carbonyl groups derivatization like the reaction with DNPH. This method is nonspecific for lipoxidation and does not detect all lipoxidation types but detects those only which preserve the carbonyl group after lipid addition. The detection of dinitrophenylhydrazone (formed after DNPH reaction with carbonyl groups) spectrophotometrically or using antibodies (against dinitrophenyl group) makes the applications of this technique broad. The same detection could also be done through biotinyl hydrazide derivatization that undergoes reaction with carbonyl groups and adds a biotin group to modified residues to enable avidin-based affinity detection or enrichment approaches. Facts suggest various antibodies can directly detect adducts of lipid peroxidation products with proteins. For example: antibodies against various HNE adducts are selective towards histidine or cysteine adducts and antibodies against adducts of MDA–lysine, oxidized phospholipid-protein and ACR–protein [275].

The antibodies against 15d-PGJ2 help in intracellular detection of PG (using immunohistochemistry) [276] and protein adducts of this lipid using ELISA [277]. Studies suggest antibodies to detect numerous unusual protein modifications with hydroperoxide-derived product which offers amide-type lipid-lysine adducts (for example HEL). In all antibody-based methods, by assuming optimal specificity of the antibody, information about modified proteins may be acquired, but information about modification sites cannot be acquired unless an antibody technique is combined with other techniques.

### 6.5. Validation Strategies

Lipoxidation targets detection and identification guides the extent of modification in biosamples and might also locate the potential targets for pathophysiological or therapeutic attention. Generally lipoxidation is non-random and occurs at specific protein residues [270].

Modified protein residues are strong nucleophiles and located in environments which favour docking or interactions with reactive lipid species [278,279]. As modified residues participate in catalysis or interact with some other proteins, it becomes important to validate the modification and assess the functional consequences of lipoxidation to define its contribution in the complete effect of lipid electrophiles. To validate lipoxidation numerous combinations of strategies are applied. Ideally a protein can be identified through a sample of a pathophysiologically appropriate model wherein the modification due to endogenously formed RCS can be detected and the site for addition can be identified using MS methods. Although, with such a method the function assessment of modification is possible, however in several methods the protein modification is done using exogenous lipid electrophiles (tagged or unmodified), but overall, under such conditions validation is obligatory. Figure 5 presents some current methods used to validate lipoxidation targets. When an electrophile is tagged, it becomes imperative to check the results using lead compounds. This is because a tag may sterically hinder interaction with the target of interest or other structures resulting in indirect effects [280]. However, the distinctive properties of untagged and tagged lipid electrophiles may be used to deduce the functional or structural consequences of modification [281].

To assess protein lipoxidation in low abundance modified proteins or peptides, modified protein immunoprecipitation or affinity-based purification methods (like: avidin–biotin interaction) are highly beneficial [245,282]. There exist some important methods to identify and characterize lipoxidation targets (the cytoskeletal protein vimentin), and to perform functional characterization of modification (Table 1) [220,283]. For functional characterization it is important to understand that oxidized lipids bind to several targets in the cell and the effect occurs due to the complex interactions of various modifications.

As discussed, earlier OS is linked with modifications that might coexist or compete with lipoxidation and complicate the structural and functional consequences. This is applicable to cysteine residue modification that apart from lipoxidation by several RCS, might also be targeted by oxidative and nitrosative modification of various structures (like cysteinylation, sulfenylation, glutathionylation, nitrosylation, etc.). As some studies suggest a relationship between lipoxidation and oxidative modification [270,285] the utmost care must be taken while allocating the function role of protein lipoxidation through specific groups without previous assessment of the overall oxidation state of proteins along with the cysteine residues.

Additionally, based on the target and structure of adduct moieties the lipoxidation may be activated or inhibited. The Ras protein modification may have dissimilar consequences on subcellular localizing and activating platforms. On one hand the little hydrophobic compounds-based modification supports localization and signalling from the Golgi apparatus. On other hand fatty acid (palmitoylation)-based modification or cyPGorcGMP addition to C-terminal cysteine supports localization and signalling from plasma membranes [270].

Application of labeled electrophile compounds together with markers (unique to distinct sub-cellular compartments) using fluorescence microscopy reveals information about specific targets in well-defined compartments. Although, in the case of a large number of cell targets for lipoxidation, colocalization findings reveal information about adducts’ subcellular localization, however this is insufficient for target validation. While considering the signals versus background effects, the amount of modified target becomes an important aspect as it may have distinct implications. Activation of little amounts such as 1% of active protein might cause a significant action, whereas inhibition of a large amount such as 10% of active protein might be insufficient. To correlate the target modification and function effect, requires mutant construct and model systems wherein target levels may be modulated. The cases wherein the lipoxidation ocurrs on a specific residue in one protein, the mutation of such a residue may occasionally block the modification and protect from the functional changes, for example vimentin (intermediate filament protein) and transcription factor NF-κB [261,280].

In cell systems, when an in vitro mutation effect can be monitored while working on purified proteins, then it becomes important to work with models which don’t express the wild type form. In the case of vimentin the transfection of cells which are not able to express the endogenous protein, allows exploration of homogeneous protein constructs/variant responses to lipid electrophiles [286]. Studies show that target residue mutations (like Cys328), rescue mutant proteins from structural derangement caused by lipid electrophiles against HNE [278]. Thereby this residue performs as a sensor to such stress either by direct modification through HNE or through RCS formation during HNE treatment. Sometimes mutations of target residues become detrimental to protein functioning and create more difficulties in the assessment of lipoxidation functional results. For example, actin is targeted by different electrophiles at the Cys374 residue, that is located on the interface between actin monomers, so its mutation per se might induce altered microfilament patterns [287].

Sometimes the function of a specific protein could be freed through mutation of a target residue. However the general outcome of cells could be further worsened when the mutated site acts as a decoy to scavenge the RCS. One must note that lipid electrophiles are vital regulators of gene expression, so protein modification with altered gene expression may result in functional effects that might amplify, or counteract the electrophiles’ effect [264,268]. Hence target modification can be confirmed based on the detection of peptide on protein adducts, whereas the functional confirmation will be based on combined strategies that involve modified residue mutations or expression level alterations.

## 7. Conclusions

Significant evidence suggests that carbonyl stress contributes to the progress of several oxidant stress-associated diseases. Promotion of carbonyl metabolism through manipulation of enzyme levels is a potential avenue in the development of novel therapeutics. CME inducers may be derived from natural compounds (through known mechanisms) or synthetic analogues (conveying similar properties). RCS scavenging exhibits significant effects in several animal models for different diseases, supporting to the notion that RCS contributes to the pathogenesis of various diseases and the scavenging of RCS could offer high clinical benefits. There are several natural and synthetic compounds that not only provide protection against carbonyl stress, but also safeguards against OS that initiate carbonyl production, although just a few RCS scavengers like MESNA and amifostine have become widely used as adjunctive therapy for chemotherapy. As several RCS species are poorly scavenged by MESNA, there is a high need for the development of other RCS scavengers.

RCS scavengers like carnosine analogues, 2-aminomethylphenols and their next generation analogues in the current scenario greatly demand improvement of their pharmacokinetic parameters and safety profiles. This is needed to fully evaluate the potential of RCS scavengers to improve human health and ameliorate chronic diseases. Nanotechnology, including nanoparticles and nanocarriers, gives hope for RCS scavengers to overcome their pharmacokinetic property problems (limited bioavailability and bioefficacy). Protein lipoxidation mechanisms offer high recognition in regulation of protein functioning in health and disease. Given the wide structural variability and complexity of these posttranslational modifications, MS-based methods are essential for their characterization. Therefore, functional assessment of modification requires integrated approaches that consider other potential modifications. As lipoxidation may cause interactions with other modifications such as oxidative modifications or adduct formation with drugs in therapeutic regimes, complex patterns may be created whose characterization requires more sensitive and specific methods. Although in vitro experiments utilizing labeled or tag-free oxidized lipids determine the potential modification sites of these compounds, however further study of protein modification in biological samples by endogenously-formed species in vivo and their function relevance will be the key for a comprehensive understanding of lipids’ role in pathophysiological scenarios.

In the current scenario several treatment approaches are available to combat RCS-associated chronic diseases. The present study recommends that prior to developing ideal therapeutics for RCS- related chronic diseases, it is essential to understand the roles of RCS in cellular physiology and the progression of various chronic diseases, along with bioavailability/bioefficacy, safety profile, and standardization (identification and validation) of therapeutics for RCS management.

## Figures and Tables

**Figure 1 antioxidants-09-01075-f001:**
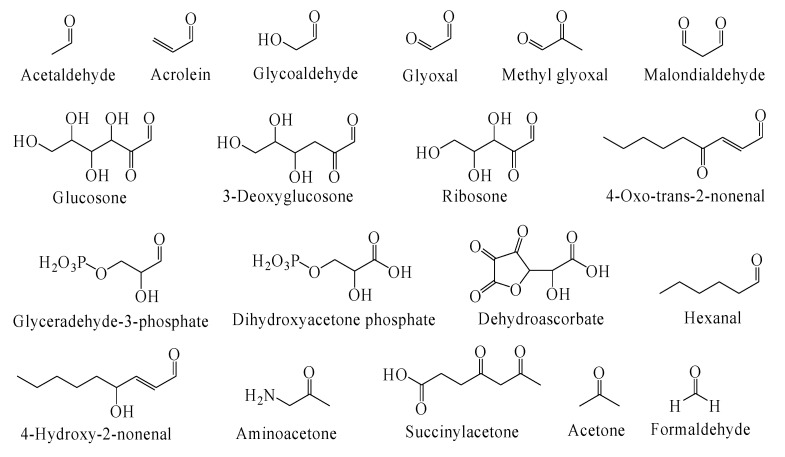
Chemical structure of commonly found RCS.

**Figure 2 antioxidants-09-01075-f002:**
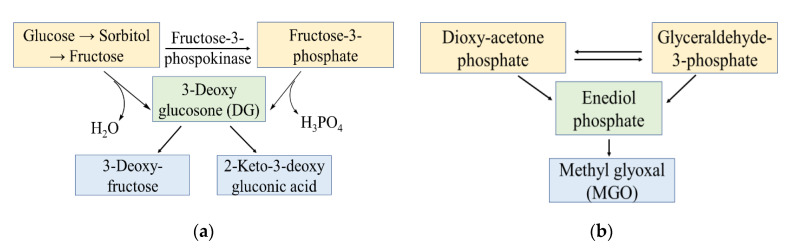
The RCS generation via polyol pathway (**a**); and glycolysis pathway (**b**).

**Figure 3 antioxidants-09-01075-f003:**
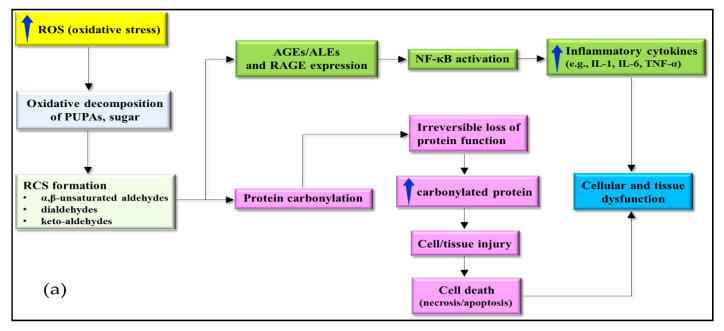
Detrimental and beneficial effects of RCS (**a**); Cytotoxic effects of RCS leading to cellular or tissue dysfunction (**b**) [90,93,94].

**Figure 4 antioxidants-09-01075-f004:**
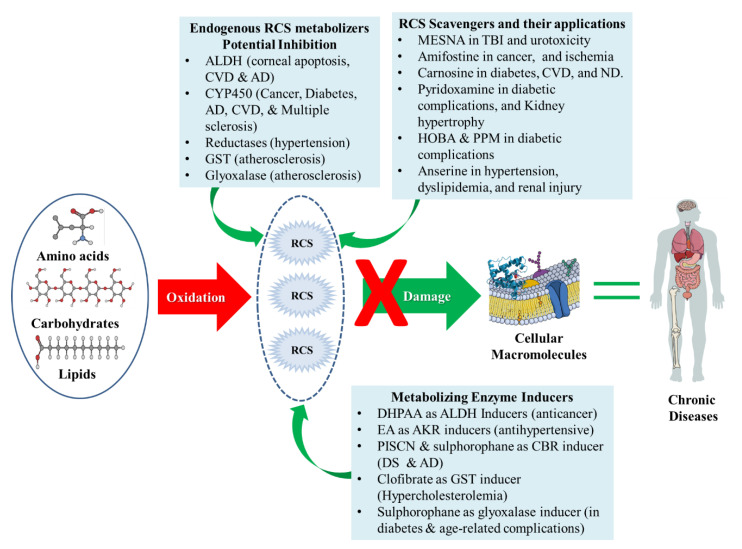
Approaches to counteract RCS at different levels.

**Figure 5 antioxidants-09-01075-f005:**
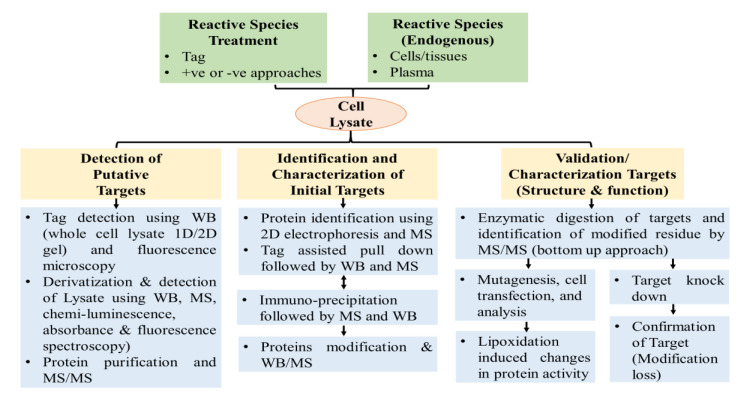
Methods for validation of lipoxidation targets.

**Table 1 antioxidants-09-01075-t001:** Combination of approaches to characterize vimentin lipoxidation and assessment of consequences of vimentin lipoxidation.

Activity	Approach	Reference
**Identification and Characterization of Vimentin**		
Identification as putative target for lipoxidation	15d-PGJ2B→Electrophoresis → WB → Detection of biotin → MALDI-TOF-MS	Stamatakis et al. [284]
	HNE → Electrophoresis → Anti-HNE-WB → Nano-LC-MALDI-MS/MS	Chavez et al. [283]
	Cell Senescence→Electrophoresis → Anti-HNE-WB → MALDI-TOF-MS	Aldini et al. [220]
Confirmation of lipoxidation site in cell	cyPG(biotinylated) → Immunoprecipitation → Detection Biotin	Aldini et al. [220]
	cyPG(biotinylation) → Immunoprecipitation → SDSPAGE → LCMS/MS	Gharbi S et al. [282]
	Carbonyl derivatization → Avidin enrichment → MS/MS	Chavez et al. [283]
Validation of lipoxidation site (Cys328)	Transfection → cyPG(biotinylation) → Avidin enrichment → SDSPAGE → WB	Gharbi et al. [275]
	Carbonyl derivatization → Avidin enrichment → MS/MS	Chavez et al. [283]
Analysis of lipoxidation in vitro	cyPG(biotinylation) → SDSPAGE → Biotin Detection	Gharbi et al. [282]
	cyPG(biotinylation)/HNE → SDSPAGE → Biotin Detection/Anti-HNEWB	Aldini et al. [220]
Confirmation of lipoxidation site in vitro (Cys328)	cyPG(biotinylation) → Digestion and Avidin enrichment → MALDI-TOF-TOF MS/MS	Gharbi et al. [282]
	In vitro competition assays	Aldini et al. [220]
**Functional Assessment of Vimentin Lipoxidation**		
In vitro assessment of lipoxidation: Filaments derangement	HNE → Polymerization (NaCl Induced) → Filaments electron microscopy	Aldini et al. [220]
In cell assessment of lipoxidation (Condensation of network by HNE treatment)	Transfection of cells (expressing vimentin) with GFP → HNE treatment → Confocal microscopy	Aldini et al. [220]
Cys328 in function consequences of vimentin lipoxidation: network condensation and filaments preservation in vimentin Cys328ser	Transfection of vimentin or Cys328ser in cell (vimentin deficient) → HNE treatment → Immuno-fluorescence → Confocal microscopy	Aldini et al. [220]

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
