# Peer review of "A Comprehensive Review on Source, Types, Effects, Nanotechnology, Detection, and Therapeutic Management of Reactive Carbonyl Species Associated with Various Chronic Diseases"

_antioxidants, 2020, doi:10.3390/antiox9111075_

Round 1

Reviewer 1 Report

This manuscript adress to an important topics such as the role of carbonyl species in chronic diseases. Here is still some points to be improved:

  1. an update of recent findings about the role of Cyp450 and MDA/GSH in signalling patway of chronic diseases (ex stroke, multiple sclerosis etc)
  2. a critical discussion about available inducers in different cronic diseases.
  3. Some minor English correction are necessary

Author Response

REVIEWERS COMMENTS AND JUSTIFCATION

Authors are sincerely thankful to the reviewer’s comments, as modification of manuscript based on reviewers comment will certainly enhance the quality of manuscript and citations.

This manuscript address to an important topic such as the role of carbonyl species in chronic diseases. Here is still some points to be improved:

1. an update of recent findings about the role of Cyp450 and MDA/GSH in signalling patway of chronic diseases (ex stroke, multiple sclerosis etc)

As per suggestion the recent findings about the role of Cyp450 and MDA/GSH in signalling patway of chronic diseases (ex stroke, multiple sclerosis etc)

 2. a critical discussion about available inducers in different chronic diseases. (both)

 As suggested a critical discussion about available inducers in different chronic diseases is incorporated in the relevant sections of the manuscript.

3. Some minor English correction are necessary

 The whole manuscript is rechecked for English corrections.

The changes made in the manuscript are highlighted with red colour font.

Apart from it some more references are added to support the incorporated text. So, the sequence of change in reference no. is highlighted with yellow colour.

Reviewer 2 Report

Congratulations, this is an excellent manuscript providing a good overview on current literature.

My only concerns relate to the title you have chosen: In your manuscript you are focussing on detection methods etc, rather than medical aspects. Due to your title I would have expected more detailed comments on the biochemical basis of diseases.

Authors have correctly pointed out that homeostasis at cellular and tissue level requires control of reactive carbonyl species as well as reactive oxygen species and reactive nitrogen species. Overproduction of any of these compounds will cause damage and can initiate diseases as exemplary listed in the review. But, because of their messenger functions, complete deletion of the compounds will cause diseases as well. Current literature describing this complex situation is listed, thus providing an overview on relevant observations. In separate chapters sources, actions and scavenging of reactive carbonyl species is described. This allows a detailed understanding of methods of reactive carbonyl detection as well as therapeutic treatment approaches.

Author Response

REVIEWERS COMMENTS AND JUSTIFCATION

Authors are sincerely thankful to the reviewer’s comments, as modification of manuscript based on reviewers comment will certainly enhance the quality of manuscript and citations.

Comment 1:

Congratulations, this is an excellent manuscript providing a good overview on current literature. My only concerns relate to the title you have chosen: In your manuscript you are focussing on detection methods etc, rather than medical aspects. Due to your title I would have expected more detailed comments on the biochemical basis of diseases.

As per suggestion the title is modified as “A COMPREHENSIVE REVIEW ON SOURCE, TYPES, EFFECTS, NANOTECHNOLOGY, DETECTION, AND THERAPEUTIC MANAGEMENT OF REACTIVE CARBONYL SPECIES ASSOCIATED WITH VARIOUS CHRONIC DISEASES”.

By modification of this title, justifies the focus on RCS detection methods.

Authors have correctly pointed out that homeostasis at cellular and tissue level requires control of reactive carbonyl species as well as reactive oxygen species and reactive nitrogen species. Overproduction of any of these compounds will cause damage and can initiate diseases as exemplary listed in the review. But, because of their messenger functions, complete deletion of the compounds will cause diseases as well. Current literature describing this complex situation is listed, thus providing an overview on relevant observations. In separate chapters sources, actions and scavenging of reactive carbonyl species is described. This allows a detailed understanding of methods of reactive carbonyl detection as well as therapeutic treatment approaches.

Authors are thankful to the reviewer for the motivational and kind words.

The changes made in the manuscript are highlighted with red colour font.

Apart from it some more references are added to support the incorporated text. So, the sequence of change in reference no. is highlighted with yellow colour.

Reviewer 3 Report

The manuscript entitled “Role of reactive carbonyl species in progression of chronic diseases and treatment approach” is a very comprehensive study on various aspects of RCS. Although the Authors analyzed many mechanisms related to RCS and their removal in the system, the study lacks specific examples of possible interventions related to chronic diseases and treatment approach.

The title indicates that we should expect such an approach to the presented information. The available literature indicates the existence of different RCS and describes them quite well. Perhaps instead of characterizing individual RCSs, a chapter could be considered to address the reader's curiosity in the context of the various cellular / systemic processes involved in the pathogenesis of disease and that may be possible therapeutic targets. It seems that this is the main message of the work, and not clear in the sheer volume of information presented. I suggest that you consider such changes to highlight the message of the next RCS review.

In my opinion, it is also worth considering the need to present the methods of RCS research in detail. You can consider it as a completely different study, mainly of a methodological nature.

Other considerations include the following:

  1. Citation or information that it refers to the text should be added to Figure 3.
  2. The title "RCS management" does not imply the content of the section. This should be changed.
  3. As a large part of the work is the chapter "RCS management", I suggest adding an appropriate figure illustrating the types of possibilities to counteract RCS at different levels, with specific examples.

Author Response

REVIEWERS COMMENTS AND JUSTIFCATION

Authors are sincerely thankful to the reviewer’s comments, as modification of manuscript based on reviewers comment will certainly enhance the quality of manuscript and citations.

Comment 1:

The manuscript entitled “Role of reactive carbonyl species in progression of chronic diseases and treatment approach” is a very comprehensive study on various aspects of RCS. Although the Authors analyzed many mechanisms related to RCS and their removal in the system, the study lacks specific examples of possible interventions related to chronic diseases and treatment approach

As suggested some more examples of possible interventions related to chronic diseases and treatment approach are further incorporated in the manuscript.

Comment 2:

The title indicates that we should expect such an approach to the presented information. The available literature indicates the existence of different RCS and describes them quite well. Perhaps instead of characterizing individual RCSs, a chapter could be considered to address the reader's curiosity in the context of the various cellular / systemic processes involved in the pathogenesis of disease and that may be possible therapeutic targets. It seems that this is the main message of the work, and not clear in the sheer volume of information presented. I suggest that you consider such changes to highlight the message of the next RCS review.

As per suggestion we have modified the titled as “A COMPREHENSIVE REVIEW ON SOURCE, TYPES, EFFECTS, NANOTECHNOLOGY, DETECTION, AND THERAPEUTIC MANAGEMENT OF REACTIVE CARBONYL SPECIES ASSOCIATED WITH VARIOUS CHRONIC DISEASES”. 

Modification of this title justifies the presentation of characterization of RCS and other appropriate contents in this manuscript.

Comment 3:

In my opinion, it is also worth considering the need to present the methods of RCS research in detail. You can consider it as a completely different study, mainly of a methodological nature. 

As per the suggestion, the need to present the methods of RCS research in detail is now included in the section 1 and 6.

Comment 4:

Other considerations include the following:

1. Citation or information that it refers to the text should be added to Figure 3. 

 As suggested citation or information that it refers to the text is added to Figure 3

2. The title "RCS management" does not imply the content of the section. This should be changed.

As per suggestion the title is modified as “A COMPREHENSIVE REVIEW ON SOURCE, TYPES, EFFECTS, NANOTECHNOLOGY, DETECTION, AND THERAPEUTIC MANAGEMENT OF REACTIVE CARBONYL SPECIES ASSOCIATED WITH VARIOUS CHRONIC DISEASES”.

3. As a large part of the work is the chapter "RCS management", I suggest adding an appropriate figure illustrating the types of possibilities to counteract RCS at different levels, with specific examples.

As per suggestion a new figure is incorporated illustrating the types of possibilities to counteract RCS at different levels, with specific examples.

The changes made in the manuscript are highlighted with red colour font.

Apart from it some more references are added to support the incorporated text. So, the sequence of change in reference no. is highlighted with yellow colour.